# Improving Barely Supervised Learning by Discriminating Unlabeled Data with Super-Class

**Guan Gui**
Nanjing University
`guiguan@smail.nju.edu.cn`

**Zhen Zhao**
University of Sydney
`zhen.zhao@sydney.edu.au`

**Lei Qi**
Southeast University
`qilei@seu.edu.cn`

**Luping Zhou**
University of Sydney
`luping.zhou@sydney.edu.au`

**Lei Wang**
University of Wollongong
`leiw@uow.edu.au`

**Yinghuan Shi**[*][†]
Nanjing University
`syh@nju.edu.cn`

## Abstract

In semi-supervised learning (SSL), a common practice is to learn consistent information from unlabeled data and discriminative information from labeled data to ensure both the immutability and the separability of the classification model. Existing SSL methods suffer from failures in barely-supervised learning (BSL), where only one or two labels per class are available, as the insufficient labels cause the discriminative information to be difficult or even infeasible to learn. To bridge this gap, we investigate a simple yet effective way to leverage unlabeled data for discriminative learning, and propose a novel discriminative information learning module to benefit model training. Specifically, we formulate the learning objective of discriminative information at the super-class level and dynamically assign different categories into different super-classes based on model performance improvement. On top of this on-the-fly process, we further propose a distribution-based loss to learn discriminative information by utilizing the similarity between samples and super-classes. It encourages the unlabeled data to stay closer to the distribution of their corresponding super-class than those of others. Such a constraint is softer than the direct assignment of pseudo labels, while the latter could be very noisy in BSL. We compare our method with state-of-the-art SSL and BSL methods through extensive experiments on standard SSL benchmarks. Our method can achieve superior results, *e.g.*, an average accuracy of 76.76% on CIFAR-10 with merely 1 label per class. The code is available at `https://github.com/GuanGui-nju/SCMatch`.

## 1 Introduction

As a paradigm to reduce the dependency on a large amount of labeled data, semi-supervised learning (SSL) has been widely concerned and utilized [1, 2]. Although existing advanced SSL methods [3, 4, 5, 6] could achieve outstanding performance even with less than 1% labels on several datasets (*e.g.*, CIFAR-10), the labeling process could still be lengthy, especially when there is a large number of object categories, which may preclude the deployment of SSL model in those applications. To tackle this challenge, barely-supervised learning (BSL), a novel paradigm with rising interest [3, 7], has been proposed recently to explore whether the model can be trained with the extremely scarce label, *e.g.*, only one label per class.

---

[*]Corresponding author.

[†]Guan Gui, Yinghuan Shi are with the National Key Laboratory for Novel Software Technology and the National Institute of Healthcare Data Science, Nanjing University.

36th Conference on Neural Information Processing Systems (NeurIPS 2022).

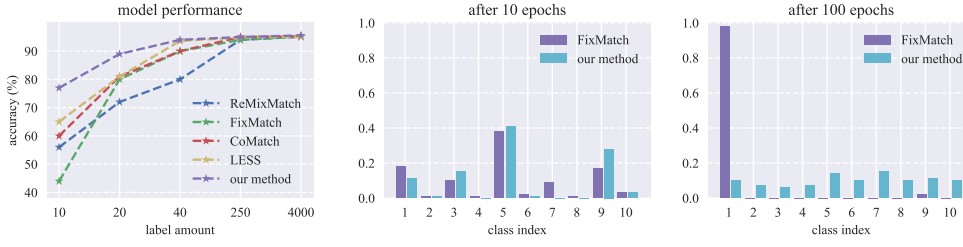

(a) performance of SSL methods       (b) predicted class distribution comparison

Figure 1: We conduct experiments on CIFAR-10 to investigate the test performance of different SSL methods with few labels. (a). Performance comparisons with different amounts of labeled data. (b). The predicted class distributions of FixMatch and our method after training the model for 10 and 100 epochs, respectively. Due to the lack of guidance from discriminative information, FixMatch tends to predict all samples as the same class, while our method alleviates this problem through our proposed discriminative information learning module.

Unfortunately, current state-of-the-art SSL methods cannot well address the label-scarce challenge in BSL. As shown in Figure 1(a), many recent SSL methods can achieve auspicious performance when sufficient labeled data are provided, *e.g.*, higher than 90% accuracy with more than 40 labels on CIFAR-10. However, these methods will suffer severe performance degeneration when the label amount is reduced. For example, when only 10 labels are available on CIFAR-10, the test accuracy of FixMatch will drop sharply by more than 45% compared to that of 40 labels. In order to explore the reasons for such performance dropping, we then track the predicted class distribution of FixMatch during the training process. As shown in Figure 1(b), FixMatch will end with the model collapse after training for 100 epochs, *i.e.*, the model completely cannot distinguish different samples, and all samples are predicted as a single class.

To analyze the reasons for above phenomenon, we prefer to investigate classification models in term of *separability* and *immutability*. Here immutability refers to the capacity of the model to be robust to perturbations. It can be mathematically expressed as $\arg\max p_m(y|u_i) = \arg\max p_m(y|\alpha(u_i))$, where $u_i$ is a sample, $y$ is the model prediction, and $\alpha(\cdot)$ is a random perturbation. Separability, on the other hand, refers to the capacity of the model to differentiate two different categories of samples, i.e., $\arg\max p_m(y|u_i) \neq \arg\max p_m(y|u_j)$, where $u_i, u_j$ are sampled from different categories.

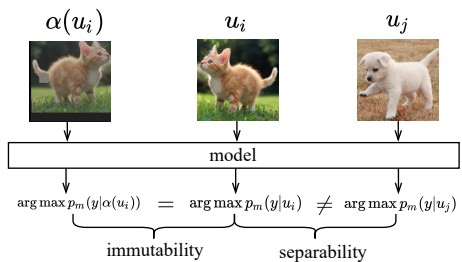

Figure 2: Examples on the immutability and the separability.

Figure 2 shows a graphic explanation of these two properties. For SSL classification models, in common practice, the immutability is often achieved by learning the consistent information of the unlabeled data, and the separability is achieved by learning the discriminative information of the labeled data. To achieve good performance, SSL models need to well balance their immutability and separability. However, such a balance is destroyed in BSL. The insufficient supervision information from the extremely scarce labels significantly damages the learning for separability, so that the model performance is dominated by the learning towards immutability. That's why the model collapse could be observed when all samples are predicted as one same class.

Motivated by these observations, in this paper, we aim to enhance the discriminative learning for the model's separability under BSL. Since the labeled data are very limited, we explore how to mine additional discriminative supervision from unlabeled data. Although without label information, the unlabeled data could still provide some "latent guidance" to complement the process of learning only from labeled data. We hereby propose a novel module to dynamically form super-classes to "roughly categorize" unlabeled samples, then the discriminative information is learned by measuring the similarity between samples and super-classes, which is realized by our newly proposed loss function on the distribution level. Furthermore, with the improvement of the model, we gradually form more

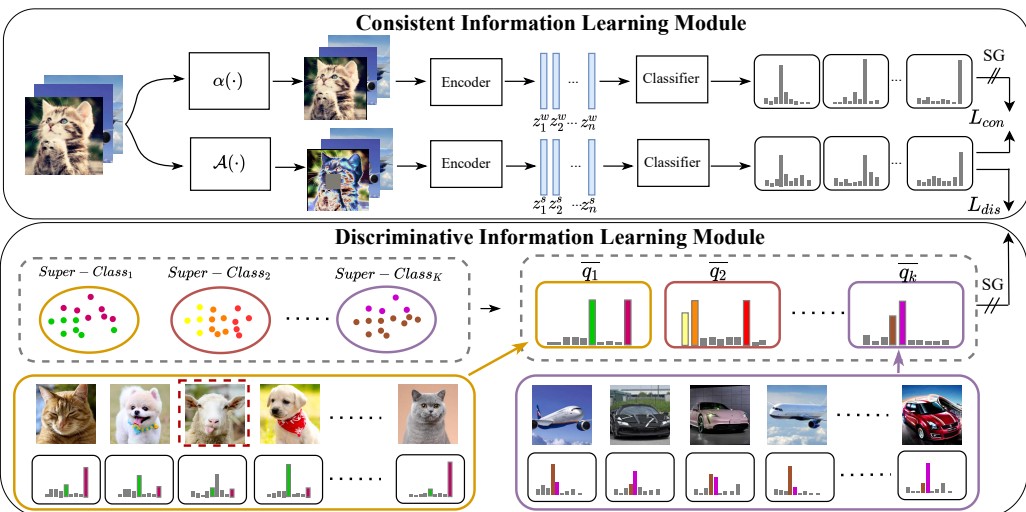

Figure 3: Overview of our method. Two different modules are constructed in our method to learn consistent information and discriminative information, respectively. Each sample $u_i$ has two variants, processed by the weak augmentation $\alpha$ and strong augmentation $\mathcal{A}$. In the consistent information learning module, $p_m(y|\alpha(u_i))$ with confidence above the threshold are then used as the training targets for the corresponding strongly-augmented variants. In the discriminative learning module, these features $z_i^w$ are clustered into $K$ super-classes. The super-class distribution $\overline{q_k}$ for the $k$-th super-class is obtained by calculating the average pseudo-labels over unlabeled data gathered in this cluster. We construct a contrastive-like loss to use these super-class distributions as guidance to train the model on the strongly-augmented variants. Specifically, the model's prediction on $\mathcal{A}(u_i)$ is encouraged to be closer to its super-class distribution $\overline{q_k}$ with $u_i \in C_k$ than to other super-class distributions.

super-classes for finer categorization of unlabeled samples, aiming to provide more fine-grained discriminative information to guide the model training.

In a nutshell, our proposed method is a simple yet effective way to mine discriminative information from unlabeled data. Compared to directly assigning pseudo labels to each sample [3, 4], which could be very noisy in BSL, learning the similarity between super-classes and samples is the softer guidance, thus reducing the error risk of pseudo labels. We evaluate our method on CIFAR-10, CIFAR-100, and STL-10, showing that our method outperforms all other SSL and BSL methods by a large margin. For example, only using one label per class on CIFAR-10, our method successfully avoided the occurrence of model collapse and achieved an accuracy of 76.76% with a variance of 6.78%.

## 2 Method

Similar to the setting of SSL, a labeled set $\mathcal{X}$ and an unlabeled set $\mathcal{U}$ are also given in BSL. $\mathcal{X} = \{(x_1, y_1), (x_2, y_2), \ldots, (x_n, y_n)\}$, where $y_i$ denotes the label of the $i$-th labeled sample $x_i$. Each sample is classified into one of $n_k$ classes denoted as $\{c_1, c_2, \ldots, c_{n_k}\}$. $\mathcal{U} = \{u_1, u_2, \ldots, u_n\}$, where $u_i$ denotes $i$-th unlabeled sample, and typically $|\mathcal{X}| \ll |\mathcal{U}|$. In BSL, a more challenging setting is considered, $|\mathcal{X}| < 4n_k$, where only few labeled data are available. In the implementation, the samples are provided on a per batch basis, with a batch of labeled data $B_x$ and unlabeled data $B_u$. As discussed before, the key to BSL lies in training a robust and stable model by efficiently leveraging the unlabeled data together with such scarce labeled data.

Unlike recent state-of-the-art SSL methods that only encourage consistency regularization on unlabeled data, our method aims to learn consistent and discriminative information from the unlabeled data simultaneously. As shown in Figure 3, we construct two modules to leverage unlabeled data accordingly, i.e., the consistent information learning module and the discriminative information learning module. In the consistent information learning module, we learn the information from the

samples and their corresponding augmented versions, like [3]. While in the discriminative information learning module, we develops the super-class distributions by clustering unlabeled samples within a mini-batch and then uses them to minimize a novel distribution loss on unlabeled samples.

## 2.1 Consistent information learning module

Like most consistency-based SSL methods, we encourage the model to output the same predictions on two differently-augmented versions of the same sample. Specifically, we produce pseudo labels on weakly-augmented samples and use them as training targets for their corresponding strongly-augmented variants. Of them, the weak augmentation $\alpha(\cdot)$ includes standard flip and shift operations, while the strong augmentation strategy $\mathcal{A}(\cdot)$ consists of RandAugment [8] and CutOut [9]. Formally, this consistency-based unsupervised loss $L_{con}$ is defined as,

$$L_{con} = \frac{1}{|B_u|} \sum_{i=1}^{|B_u|} \mathbb{1}(\max(p_m(y|\alpha(u_i))) \geq \tau_1) H(p_m(y|\alpha(u_i)), p_m(y|\mathcal{A}(u_i))) \qquad (1)$$

where $H(p_1, p_2)$ denotes the standard cross entropy between $p_1$ and $p_2$, and $\tau_1$ is a pre-defined threshold to retain only high-confidence pseudo labels. As discussed in [3], $\tau_1$ is commonly set as a high value to alleviate the confirmation bias in SSL.

## 2.2 Discriminative information learning module

In addition to relying on labeled data to learn discriminative information, we propose a novel module, an on-the-fly learning process to first form super-classes and then exploit the similarity between super-classes and samples to improve the model's separability.

One of the most intuitive ways to explore discriminative information is to generate class information for unlabeled data by clustering in the feature space. Ideally, samples of the same category will form a separate cluster so that the model can discriminate the samples from all other $n_k - 1$ clusters of samples. However, forming such fine-grained clusters carries a considerable risk of errors, especially for tasks with a large number of object categories. What's worse, in the early training stage, due to the weak feature extraction ability, the model inevitably produces wrong discriminative information, resulting in severe accumulated errors. To properly explore the discriminative information for unlabeled data, we propose the following designs,

- First, instead of fine-grained clusters, we simplify the clustering task by allowing a cluster to contain multiple categories, i.e., a super-class cluster. In this way, the discriminative information is relatively weakened but more robust to clustering errors.

- Second, it can still be noisy to adopt the super-class label as training targets for unlabeled data. Therefore, we tend to utilize the similarity between each sample and the super-classes rather than explicitly assign the training targets for unlabeled data. Concretely, our method encourages the unlabeled samples to stay closer to the predicted class probability distribution of their corresponding super-class than those of others. Such a smoothing way can better tolerate the inaccurate prediction of a single sample as well as potential clustering errors.

- Third, although the discriminative information provided by the coarse-grained clusters is robust, it will be insufficient when the model's separability is improved. Thus we propose the progressive construction of super-classes to gradually increase the clustering number so that our discriminative information learning module can adapt to the model evolution during the training process. When the cluster number is small, each super-class provides more moderate discriminative information, called a low-level super-class. In contrast, a large cluster number can enforce each super-class to abstract more concrete information, and we call it a high-level super-class.

**Super-class representation**

As shown in Figure 3, we employ standard K-Means on these features $z_i^w$ of weakly-augmented samples within a mini-batch. With a given target number $K$ of super-classes, these features are gathered into $K$ clusters, and each cluster is denoted by $C_k, k = 1, 2, \ldots, K$. Each super-class can then be represented by the mean distribution of all the samples it contains. Given unlabeled sample

$u_i$, and its predicted class probability distribution $p_m(y|\alpha(u_i))$ and $p_m(y|\mathcal{A}(u_i))$, the super-class distribution $\overline{q_k}$ for each super-class $C_k$ can be calculated by,

$$\overline{q_k} = \frac{1}{|C_k|} \sum_{i=1}^{|C_k|} p_m(y|\alpha(u_i)), \quad \text{with } u_i \in C_k \tag{2}$$

In this way, the super-class distribution can represent the distribution characteristics of the categories it contains so that it can be well discriminated from other super-classes. As shown in the lower half of Figure 3, in the automobile-and-airplane super-class, it is possibly not easy to determine the exact category for a single sample. However, we can find that the sample in this super-class should be closer to the super-class distribution of the automobile-and-airplane super-class compared to those of other super-classes. Additionally, the super-class is more robust to the noisy samples. For samples likely to be misclassified (*e.g.*, samples inside the dashed box), their negative impact on the super-class distribution is well suppressed by other correctly classified samples.

**Discriminative distribution loss**

To distinguish the sample from other super-classes, this sample is supposed to be more similar to its corresponding super-class on distribution. Inspired by [10, 11], we design a contrastive-like distribution loss to distinguish the sample from other super-classes. Formally, this auxiliary distribution loss is,

$$L_{dis} = -\frac{1}{|B_u|} \sum_{i=1}^{|B_u|} \mathbb{1}(\max(p_m(y|\alpha(u_i))) \geq \tau_2) \log \frac{\exp(p_m(y|\mathcal{A}(u_i)) \cdot \overline{q_k}/T)}{\sum_{j=1}^{K} \exp(p_m(y|\mathcal{A}(u_i)) \cdot \overline{q_j}/T)} \tag{3}$$

where $T$ is a common temperature parameter, and $u_i$ is assigned to super-class $C_k$. Like in Equation 1, we adopt a parameter $\tau_2$ to control the learned unlabeled data. As mentioned before, the similarity between samples and super-classes is a weak constraint, so we are conditioned to use a lower threshold to learn more samples. We provide an empirical value via extensive ablation studies. Notice that we compute gradients only on strongly-augmented samples.

**Progressive super-class construction**

Although small $K$ reduces the clustering error, it comes at the cost that the learned discriminant information is limited. Assuming the extreme case, when $K = 1$, the amount of information is 0 because all samples belong to one super-class, the model will not discriminate against any samples.

With this point of view, we propose the progressive construction of super-class to adapt to the model evolution during training. That is, when the model is not well trained at the beginning, we use a small $K$ to form the coarser super-classes to ease the clustering task and thus attain relatively reliable discriminative guidance. When the model is better trained, to avoid the training of the model being stagnant due to the limitation of discriminative information, we gradually increase $K$ to provide enhanced discriminative guidance.

In practice, a daunting challenge is that we do not know the most appropriate number of super-class for the training samples without prior knowledge. To this end, we design a dichotomous method and set the value range of $K$ by:

$$K_i \in \{2, \ldots, \lceil n_k/4 \rceil, \lceil n_k/2 \rceil, n_k\} \tag{4}$$

The above formula restricts the value of $K$ based on the principle of dichotomy so that frequent changes of $K$ can be avoided. Especially when there are many classes in the sample set (*e.g.*,100 classes on CIFAR-100), it would be tedious and pointless to learn all $K$ values. Furthermore, to ensure that the clustering task with different $K$ can be performed for a certain period, we adopt a linear-step growth strategy to adjust $K$ dynamically:

$$K = K_i, \quad \text{if} \quad K_i \leq \frac{t}{\alpha * t_s} < K_{i+1}, \tag{5}$$

where $t$ and $t_s$ denote the value of the current iteration and the total number of iterations, respectively. $\alpha \in (0, 1)$ and it controls the growth rate of $K$. With this clustering task, $K$ super-classes are dynamically formed at each iteration.

---

**Algorithm 1** Algorithm of our method

---

**Input**: Labeled batch $B_x = \{(x_i, y_i)\}$, unlabeled batch $B_u = \{u_i\}$, weak augmentation strategy $\alpha(\cdot)$, strong augmentation strategy $\mathcal{A}(\cdot)$

**Parameter**: threshold $\tau_1, \tau_2$, temperature $T$, loss weight $\lambda_{con}, \lambda_{dis}$

1: compute $L_{sup} = \frac{1}{|B_x|} \sum_{i=1}^{|B_x|} H(p_m(y|\alpha(u_i)), y_i)$:

2: **for** $t \leftarrow 1$ to $t_s$ **do**

3:     **for** $u_i \in B_u$ **do**

4:         $z_i^w = \text{Encoder}(\alpha(u_i))$         // record features of weakly augmented samples.

5:         $p_m(y|\alpha(u_i)), p_m(y|\mathcal{A}(u_i))$     // compute prediction of $\alpha(u_i)$ and $\mathcal{A}(u_i)$.

6:     **end for**

7:     $L_{con} = \frac{1}{|B_u|} \sum_{i=1}^{|B_u|} \mathbb{1}(\max(p_m(y|\alpha(u_i))) \geq \tau_1) H(p_m(y|\alpha(u_i)), p_m(y|\mathcal{A}(u_i)))$

8:     update $K$.

9:     form super-classes by K-Means$(K, z_i^w)$.

10:    $\overline{q_k} = 1/|C_k| \sum_{i=1}^{|C_k|} p_m(y|\alpha(u_i)), \quad \forall u_i \in C_k$

11:    $L_{dis} = -\frac{1}{|B_u|} \sum_{i=1}^{|B_u|} \mathbb{1}(\max(p_m(y|\alpha(u_i))) \geq \tau_2) \log \frac{\exp(p_m(y|\mathcal{A}(u_i)) \cdot \overline{q_k}/T)}{\sum_{j=1}^{K} \exp(p_m(y|\mathcal{A}(u_i)) \cdot \overline{q_j}/T)}$

12:    minimizing the total loss $L = L_{sup} + \lambda_{con} L_{con} + \lambda_{dis} L_{dis}$.

13: **end for**

---

## 2.3 Total Loss

Similar to most SSL methods, the supervised loss for a batch of labeled data $B_x$ is obtained by a standard cross-entropy loss,

$$L_{sup} = \frac{1}{|B_x|} \sum_{i=1}^{|B_x|} H(p_m(y|\alpha(u_i)), y_i) \tag{6}$$

In summary, the total loss in our method is,

$$L = L_{sup} + \lambda_{con} L_{con} + \lambda_{dis} L_{dis} \tag{7}$$

where $\lambda_{con}$ and $\lambda_{dis}$ are the weights of $L_{con}$ and $L_{dis}$, respectively. The full algorithm is provided in Algorithm 1.

## 3 Experiments

In this section, we validate the effectiveness of our proposed method by conducting experiments on widely-used SSL benchmark datasets: CIFAR-10, CIFAR-100 [12], and STL-10 [13].

### 3.1 Implementation details

To face the challenge of BSL, we randomly sample 1 or 2 labels for each class on these data sets. We adopt "WideResNet-28-2" and "WideResNet-28-8" [14] as the backbone for CIFAR-10 and CIFAR-100, respectively, while using "ResNet18" [15] for STL-10. For the consistent information learned module, we follow the same setting with [3], where $\tau_1 = 0.95, |B_x| = 64, |B_u| = 7|B_x|$. And for the discriminative information learned module, we set $T = 1, \tau_2 = 0.8$. In addition, Since the essence of the three losses of $L_{sup}, L_{con}, L_{dis}$ is in the form of cross entropy, it's prefer to set $\lambda_{con} = \lambda_{dis} = 1$ to further reduce of hyperparameters. For CIFAR-10 and STL-10 task, we set the $K \in \{\lfloor n_k/3 \rfloor, n_k/2, n_k\} = \{3, 5, 10\}$. For CIFAR-100, considering that the samples of each cluster should be sufficient, we set the $K \in \{n_k/20, n_k/10, n_k/5\} = \{5, 10, 20\}$. In fact, the specific numerical setting of K has little effect on the model performance, and more analysis and experiments about $K$ will be discussed in the later ablation experiments.

Table 1: Performance comparisons on CIFAR-10, CIFAR-100, STL-10. Each result is reported as the average of 5 runs. The results show our method outperforms other baselines in all settings. On CIFAR-10 and STL-10 with 10 labels, our method outperforms other methods by at least 10%. For the larger dataset CIFAR-100, our method also outperforms baseline methods by at least 6%.

| | CIFAR-10 | | CIFAR-100 | | STL-10 | |
| Method | 10 labels | 20 labels | 100 labels | 200 labels | 10 labels | 20 labels |
| --- | --- | --- | --- | --- | --- | --- |
| Mean-Teacher | $15.48 \pm 3.19$ | $17.50 \pm 1.16$ | $5.17 \pm 2.52$ | $8.26 \pm 3.43$ | $11.05 \pm 6.45$ | $15.99 \pm 6.45$ |
| MixMatch | $17.18 \pm 4.45$ | $26.45 \pm 8.17$ | $12.85 \pm 2.21$ | $21.56 \pm 4.84$ | $10.94 \pm 5.18$ | $21.48 \pm 3.17$ |
| ReMixMatch | $60.29 \pm 15.20$ | $78.56 \pm 9.63$ | $26.18 \pm 3.79$ | $35.90 \pm 3.66$ | $30.86 \pm 10.80$ | $45.58 \pm 8.36$ |
| FixMatch | $44.47 \pm 24.99$ | $80.46 \pm 5.15$ | $25.49 \pm 4.37$ | $35.55 \pm 1.59$ | $25.75 \pm 8.99$ | $48.98 \pm 6.46$ |
| FixMatch (w/DA) | $67.79 \pm 15.42$ | $84.16 \pm 9.27$ | $31.10 \pm 2.29$ | $43.22 \pm 1.87$ | $42.08 \pm 6.24$ | $54.76 \pm 5.44$ |
| CoMatch | $60.79 \pm 12.42$ | $81.19 \pm 8.55$ | $27.54 \pm 4.25$ | $36.98 \pm 2.17$ | $29.11 \pm 9.31$ | $50.20 \pm 7.57$ |
| FlexMatch | $66.07 \pm 10.58$ | $85.69 \pm 6.24$ | $31.50 \pm 3.61$ | $38.05 \pm 2.66$ | $41.17 \pm 6.20$ | $54.30 \pm 5.65$ |
| SLA | $65.87 \pm 10.83$ | $81.89 \pm 6.77$ | $28.45 \pm 2.16$ | $38.65 \pm 2.67$ | $32.38 \pm 8.32$ | $47.50 \pm 6.38$ |
| LESS | $64.40 \pm 10.90$ | $81.20 \pm 5.60$ | $28.20 \pm 3.00$ | $42.50 \pm 3.20$ | $34.25 \pm 7.19$ | $48.98 \pm 5.19$ |
| our method | $\mathbf{76.76 \pm 6.78}$ | $\mathbf{88.49 \pm 3.26}$ | $\mathbf{37.50 \pm 1.72}$ | $\mathbf{45.62 \pm 1.39}$ | $\mathbf{52.51 \pm 3.20}$ | $\mathbf{57.98 \pm 3.18}$ |

The model is trained with a total of $2^{20}$ iterations, and the $K$ increased in the first 30% iterations. We use an exponential moving average with a decay rate of 0.999 to test our model and repeat the same experiment for five runs with different seeds to report the mean accuracy.

## 3.2 Baseline methods

First, FixMatch [3], Dash [16], CoMatch [5], FlexMatch [4] are the advanced semi-supervised models in recent years, and we compare these methods under the challenge of barely-supervised learning. We also use FixMatch with the distribution alignment (DA). SLA [6] and LESS [7] are the latest models on BSL, and we also use them as our comparison method. In addition to this, we also select some classical semi-supervised methods such as MeanTeacher [17], MixMatch [18] and ReMixMatch [19] for comparison.

## 3.3 Experimental results

**Performance comparisons**. In Table 1, we compare the test accuracy of our proposed method against recent SSL and BSL methods. It can be seen that our results are state-of-the-art in all settings. Especially when there is only one label per class, our method compensates for the shortage of labeled data by mining latent discriminative information from unlabeled data, thus showing enormous superiority. LESS [7], recent work on BSL, since it generates predictions for samples with low confidence and then learns more consistent information, still ignores the learning of discriminative information, it cannot solve the challenges in BSL.

On CIFAR-10 task with 10 labels, our method achieves the mean accuracy of 76.76%, which outperforms other methods by 10%. On STL-10 task with 10 labels, the recent BSL methods LESS and SLA achieve the accuracy of 34.25% and 32.38%, respectively, while our method achieves the mean accuracy of 52.51%, which improved nearly by 20%. For larger dataset CIFAR-100, our method also outperforms other methods by at least 6% when there is 1 label per class. Besides, we can see that, regardless of the dataset, the performance of our method when using only 1 label per class is close to or even exceeds the performance of other methods when using 2 labels per class. On CIFAR-100 task, LESS and SLA achieve the mean accuracy of 42.50% and 38.65% with 100 labels, while our method achieves the mean accuracy of 37.50% with half of the labels they use.

As mentioned by [3], the quality of very few labeled data will significantly affect the performance of the model. Taking the CIFAR-10 task with 10 labels as an example, the variance of advanced SSL methods and BSL methods are all more than 10%, while the variance of our method is only 6.78%. These results further illustrate that our method can alleviate the dependence on labeled data by learning discriminative information from unlabeled data.

We also find that the technique of distribution alignment, which forces the alignment of probability distributions is still an effective technique under BSL. Through the result of FixMatch (w/DA), we can see that DA successfully helped FixMatch improve its performance significantly. However, DA is a technique that relies on prior information, and our method does not rely on any prior information and can achieve better performance than it.

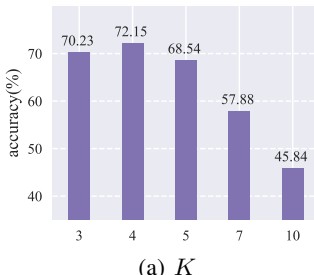
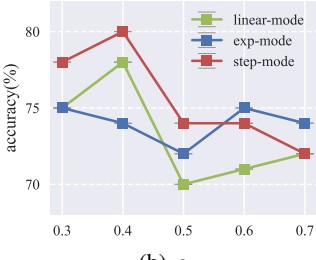
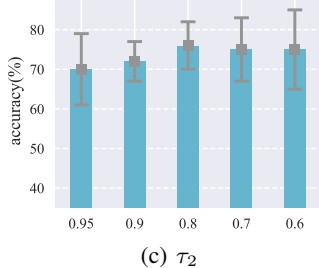

(a) $K$        (b) $\alpha$        (c) $\tau_2$

Figure 4: Abalation study on CIFAR-10 with 1 labeled sample per class. **(a)** Performance using fixed K values without dynamical clustering to form super-classes. **(b)** performance using progressive $K$ with different growth rate $\alpha$. **(c)** Confidence threshold $\tau_2$. Appropriately lowering the threshold can learn more samples, thereby helping model training.

**Stability of the model**. First of all, we discuss the phenomenon of model collapse under the BSL challenge. For a fair comparison, we use the same random seed in each trial for FixMatch and our method. Since only 1 label per class is available, this SSL method that depends on labeled data to learn discriminative information would be volatile. As shown in table 2, we can see that the per-

Table 2: Experiments on CIFAR-10 with 10 labels (1 label per class). FixMatch exhibits volatile performance, while our method greatly improves performance and is stable.

| seed | 1 | 2 | 3 | 4 | 5 |
|------|------|------|------|------|------|
| FixMatch | 19.15 | 85.11 | 52.52 | 17.09 | 48.50 |
| our method | **81.28** | **86.12** | **70.34** | **74.90** | **71.17** |

formance of FixMatch is extremely unstable, where it can achieve very high accuracy of $85.11\%$ when seed$= 3$ but obtain an extremely low accuracy of $17.09\%$ when seed$= 4$. Differently, integrating the proposed super-class distribution to provide more discriminative information, our method can successfully alleviate the model collapse: the accuracy exceeded $70\%$ in all experiments and also exceeded $80\%$ sometimes.

**Performance under SSL settings.** We also analyze our method in standard SSL settings where sufficient labeled data are provided. As shown in Table 3, we test our method on CIFAR-10 with 40, 250, and 4000 labels. It can be seen that when the number of labels increases, our method is not SOTA, but the gap with other methods is within 1%. It can be interpreted as these methods using other advanced techniques in the learning consistent information pro-

Table 3: Experiments on CIFAR-10 with more labels. When there are enough labeled data, our method performs on a par with SOTA SSL methods.

| Method | 40 labels | 250 labels | 4000 labels |
|--------|-----------|------------|-------------|
| ReMixMatch | $80.90 \pm 9.64$ | $94.56 \pm 0.05$ | $95.02 \pm 0.11$ |
| FixMatch | $90.11 \pm 3.01$ | $94.21 \pm 0.61$ | $95.91 \pm 0.02$ |
| CoMatch | $93.09 \pm 1.39$ | $95.09 \pm 0.33$ | $95.57 \pm 0.20$ |
| FlexMatch | $95.01 \pm 0.16$ | $95.20 \pm 0.06$ | $96.05 \pm 0.03$ |
| SLA | $94.83 \pm 0.32$ | $94.98 \pm 0.28$ | $95.59 \pm 0.09$ |
| LESS | $93.20 \pm 2.10$ | $95.10 \pm 0.80$ | $95.64 \pm 0.29$ |
| our method | $94.19 \pm 0.41$ | $94.54 \pm 0.27$ | $95.78 \pm 0.08$ |

cess. For example, FlexMatch [4] and SLA [6] both leverage prior knowledge of class proportions, and CoMatch [5] leverages graph-based contrastive learning, *etc.*While we study an independent module for solving BSL, so when the labels are enough, our method does not prevail. However, though our implementation is based on FixMatch [3], the results show that our method can still improve slightly when the number of labels is large.

## 3.4 Ablation study

**Performance under different strategies of $K$.** We explore the effect of $K$ for different strategies on the model: (1) fix-mode, the number of super-class remains constant during model training. (2) linear-mode, the number of super-class increases linearly during model training. (3) exp-mode, the number of super-class increases exponentially. (4) step-mode, a step-by-step jump growth based on linear-mode.

We fixed different $K$ values for experiments in terms of the fixed strategy. As shown in Figure 4(a), when $K$ is small, the model tends to outperform the larger $K$. When $K$ is large, the model in the early stage does not provide high-quality features to perform the formation of high-level super-classes, so the discriminative information learned has an extremely high risk of error, leading to the model's failure. On the other hand, when $K$ is small, our method can learn effective discriminative information from these low-level super-classes. However, as the model performance improves, this limited discriminative information provided by low-level super-classes can no longer help the model learn continuously, so the model's performance will stagnate. It is worth noting that even if we adopt the fix-mode with $K$, the model can learn a certain degree of discriminative from the super-class to face the challenge of BSL, and its performance also exceeds other methods.

Linear-mode, exp-mode, step-mode can all work well to solve the problem in the fix-mode above, while there is an additional hyperparameter to control the rate of $K$ growth in these modes. As shown in Figure 4(b), we conduct experiments for different growth rates, and it turns out that the growth rate of $K$ does not affect the performance of the model too much. In addition, since this mode can explore different levels of discriminative information, the performance is significantly better than that in the fix-mode. Although the performance of these modes is exceptionally close, we prefer step-mode as it can be more suitable for large data sets, *e.g.*, it is impractical to increase $K$ from 3 to 100 sequentially when we test on CIFAR-100.

**Performance under different $\tau_2$.** We investigate 5 different $\tau_2$ values on CIFAR-10 datasets with 10 labels. As shown in Figure 4(c), the test performance achieve the best when $\tau_2 = 0.8$. It shows that appropriately lowering the threshold can learn discriminative information from more samples, thereby helping model training. However, if the threshold $\tau_2$ is too low, the noise of the sample will increase, which is not conducive to the training of the model.

# 4 Related Work

Recent popular semi-supervised learning studies can be classified into entropy minimization (ER) based methods and consistency regularization (CR) based methods. Self training is the typical representative of ER-based methods. In these methods [1], the model is first trained on the provided labeled data and then used to generate pseudo-labels for unlabeled data. After that, such methods add these unlabeled data with high-confidence predictions into the labeled set to retrain the model, repeating this process until all unlabeled data are involved [20]. Recent studies tend to involve more advanced techniques in this framework to enhance the SSL performance. [21] introduces multiple views to provide more robust pseudo-labels. LaSSL [22] and Curriculum Labeling [23] integrate the contrastive learning and curriculum learning techniques to improve the accuracy of pseudo-labels further.

As the most widely-used and successful technique in recent SSL methods, CR is the semantics of a sample should be consistent after data perturbations [24, 17, 18, 25, 26]. FixMatch [3] combines strong augmentation technology [8, 9] and the labels of weakly augmented samples with high confidence are used to guide the learning of strong augmented samples. Although a major breakthrough has been made in conventional semi-supervised learning, it still cannot avoid model collapse. [16, 4] further dynamically adjusts the confidence threshold based on FixMatch. Although it can learn more low-confidence samples to improve the performance of the model, it cannot cope with the challenge of lack of discriminative information under BSL.

In the literature, there have been few works on barely-supervised learning. FixMatch [3] initially came up with the concept of BSL and emphasized that the quality of the labeled data played a crucial role in the test performance. Our experiment results also demonstrate that its testing results have a very high variance under the BSL settings. Recent SLA [6] also achieved better performance in BSL by formulating an optimal transportation problem between samples and labels. It introduced many extra hyper-parameters and adopted the Sinkhorn-Knopp algorithm to solve the optimization problem approximately. Differently, our method gets rid of complicated operations but can effectively improve the BSL performance. Another work [7] argues that the dilemma in BSL is that there are no pseudo-labels that can be predicted with high confidence, so online deep clustering is used to supplement the pseudo-labels predicted by the model. Although more pseudo-labels can be used, it still learns consistent information. [27] is a very recent work that uses the coarse-grained class labels to guide the SSL model. However, it requires strong prior knowledge about the class hierarchy

structure in advance; while we are faced with BSL scenarios without any prior knowledge, even the hierarchy structure may not exist.

## 5 Conclusion

In this paper, we analyze the failure of SSL methods in the face of BSL as insufficient discriminative information learning. To tackle this problem, we design a discriminative learning module to leverage unlabeled data for additional discriminative supervision. In this module, super-classes are dynamically reformed with the model training, and then the discriminative information is learned by measuring the similarity between samples and super-classes. We conduct our methods on several SSL benchmarks, and it shows that our method outperforms other methods in BSL.

## 6 Acknowledgment

This work was supported by the Science and Technology Innovation 2030 New Generation Artificial Intelligence Major Project (2021ZD0113303), the NSFC Program (62222604, 62206052, 62192783), CAAI-Huawei MindSpore Project (CAAIXSJLJJ-2021-042A), China Postdoctoral Science Foundation Project (2021M690609), Jiangsu Natural Science Foundation Project (BK20210224), and CCF-Lenovo Bule Ocean Research Fund.

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
