# OpenReview forum: "Improving Barely Supervised Learning by Discriminating Unlabeled Samples with Super-Class"
_NeurIPS.cc/2022/Conference — NeurIPS 2022 Accept_

### Official Review · Reviewer_vR2e · 2022-07-07

**Rating:** 4
**Confidence:** 4
**Soundness:** 2 fair
**Presentation:** 3 good
**Contribution:** 2 fair

**Summary:**

This paper presents a semi-supervised learning method for barely supervised setting. The core contribution is the design of discriminative information learning module with clustered super classes. The experiment on barely supervised setting demonstrates the consistently improved results on small datasets.

**Questions:**

Questions:
For eq.3, why a constative-like loss is exploited. Is simple CE loss on the super classes not working? Cosine similarity is adopted in eq.3, but for softmax probabilities.

It important to show the training speed with the proposed discriminative information learning module. Would running K-means for each training iteration slow down the training? Comparison between the training speed would be expected.

For barely supervised setting, random seed often plays important roles for the distribution of the labeled data. Can you provide more details on how the random seed is set? Is it randomly selected or with more sophisticated selection? For example, in FixMatch, they expain how they choise the 10 labels for CIFAR-10

All experiments are run with small datasets. Also, a challenging dataset SVHN is not shown. SVHN is more challenging because its noisy background. It would be interesting to see the performance on SVHN and larger datasets such as (Mini)-ImageNet.
To my understanding, for barely supervised setting, the quantity of the samples accepted for calculating the loss is especially important for early training. If a high threshold (tau_1, tau_2) is employed, the SSL algorithms would collapse to one class as shown in Fig.1 because not diverse enough samples are enrolled during training. This is also demonstrated in Fig.4(c), when tau_2 is higher, the performance drops more significantly than when tau_2 is lower. In that case, compare the enrolled rate of samples and the pseudo-label accuracy to FixMatch and FlexMatch is important to demonstrate the effectiveness of the proposed method.

The Super class concept is very similar to SimMatch (https://arxiv.org/abs/2203.06915) yet no discuss and comparision is provided.

**Limitations:**

see above

**Strengths And Weaknesses:**

Strengths:
The paper is well-written and organized, easy to follow.
The improvement on the studied settings is significant and consistent.

Weakness:
The design choice on auxiliary distribution loss (eq.3) is not shown
There is a mismatch of EMA of model parameter in main paper 0.999 (L201) versus 0.99 in appendix 0.99 (in table1)

---

> ### Author Response · Authors · 2022-08-02
> **Response to reviewer vR2e (Q5-Q6)**
>
> ## Q5: analysis on the amount and accuracy of high-confidence pseudo-labels
> As discussed in the literature, the high confidence-threshold is a simple yet effective and necessary strategy to alleviate the confirmation bias in consistency-based SSL methods. It may affect the quantity of involved unlabeled samples. However, **we argue that the reason why the model collapse occurred is not because high thresholds prevented diverse samples from being enrolled, but because that the model lacks enough discriminative guidance so that most samples were predicted to be in the same category with high confidence**, i.e., the pseudo-label accuracy of the high-confident samples is low. To further verify our claim, we provide more training details in terms of the high-confidence quantity and pseudo-label accuracy in the tables below.
>
> |#iterations|1w|2w|3w|4w|5w|6w|7w|8w|9w|10w|20w|30w|40w|50w|100w|
> |-|:-:|:-:|:-:|:-:|:-:|:-:|:-:|:-:|:-:|:-:|:-:|:-:|:-:|:-:|:-:|
> |FixMatch|0.84|0.92|0.99|0.99|0.96|0.99|0.99|0.99|0.99|0.99|0.95|0.94|0.97|0.99|0.99|
> |FlexMatch|0.86|0.84|0.90|0.86|0.82|0.84|0.86|0.84|0.90|0.90|0.92|0.91|0.91|0.94|0.95|
> |ours    |0.68|0.71|0.84|0.80|0.89|0.84|0.85|0.86|0.88|0.87|0.90|0.92|0.90|0.93|0.94|
>
>
> Table R4-b. Percentage of high-confidence pseudo-label.
>
> |#iterations|1w|2w|3w|4w|5w|6w|7w|8w|9w|10w|20w|30w|40w|50w|100w|
> |-|:-:|:-:|:-:|:-:|:-:|:-:|:-:|:-:|:-:|:-:|:-:|:-:|:-:|:-:|:-:|
> |FixMatch|0.14|0.11|0.17|0.12|0.12|0.13|0.08|0.12|0.10|0.10|0.11|0.13|0.10|0.13|0.18|
> |FlexMatch|0.20|0.32|0.35|0.29|0.31|0.31|0.32|0.32|0.35|0.53|0.68|0.69|0.68|0.69|0.68|
> |ours    |0.19|0.33|0.35|0.33|0.37|0.55|0.67|0.66|0.64|0.63|0.75|0.79|0.81|0.84|0.85|
>
> Table R4-c. Accuracy of high-confidence pseudo-label.
>
> We can clearly see that, the baseline (FixMatch) have a very high quantity of high-confidence samples above the threshold (i.e. the amount of involved unlabeled samples) under the BSL scenario, even higher than that of both FlexMatch and our method. However, its accuracy of pseudo-labels is very low. We can also have similar observation on FlexMmatch. On the contrary, exploring discriminative guidance in our methods can gradually increase the amount of high-confidence pseudo-labels and more importantly, the accuracy of pseudo-labels, resulting in higher test accuracy in the end.
>
> ## Q6: discussion and comparison on SimMatch
> Thanks for sharing this new CVPR paper. We investigate its performance under the BSL scenario as follows.
>
> ||10 labels (seed = 1)| 10 labels (seed = 2) | 20 labels (seed = 1) | 20 labels (seed = 2)
> |-|:-:|:-:|:-:|:-:|
> |SimMatch|24.28|76.37|80.69|83.77|
> |ours|81.28|86.12|88.61|87.91|
>
>
> Table R4-d. Comparison between SimMatch and our approach under BSL on CIFAR10.
>
> We can clearly observe that our methods can consistently outperform SimMatch, especially with 10 labels. On the other hand, in terms of methodological design, although both SimMatch and our work use distributional similarity, they are used from completely different perspectives and for completely different purposes.  We summarize the differences as follows.
>
> - SimMatch's similarity is computed between samples and samples, whereas our similarity is computed between samples and super-classes.
> - The aim of SimMatch is to make the similarity between the strongly augmented and weakly augmented views of the image consistent, whereas our aim is for the image to be apart from other super-classes and be close to its corresponding super-class.
> - SimMatch focuses mainly on learning more consistency information (between strongly augmented and weakly augmented views of an image). Differently, in addition to consistent information, our methods focus on learning new discriminative information (between images and super-classes).
>
> As analysed in our paper, the problem faced by the SSL models under BSL is the lack of discriminative information learning, which leads to model collapse. Such a problem also exists in SimMatch, which does not use similarity to learn discriminative information. This explains the possibility of model collapse when there are only 10 labels on CIFAR-10, with an accuracy of 24.28%. To the best of our knowledge, we are the first work to think in terms of discriminative information towards solving the BSL problem.
>
> ## Writing errors
> The EMA of model parameter is 0.999, the 0.99 in the supplementary material is an writing error, thank you for pointing it out and we will correct it in a subsequent version.

---

> ### Author Response · Authors · 2022-08-02
> **Response to reviewer vR2e (Q1-Q4)**
>
> ## Q1: about CE-loss and constative-like loss.
> |CE loss with hard label|CE loss with softmax probabilities|constative-like loss|
> |:-:|:-:|:-:|
> |17.01|27.24|81.28|
>
> Table R4-a. Accuracy achieved by CE-loss and the contrastive-like loss on CIFAR-10 with 10 labels
>
>
> As suggested, we apply CE loss on unlabeled data to enforce the prediction consistency to its super-class. As shown in Table R4-a, we test the performance on CIFAR10 using the same random seed 1. Clearly, with CE-loss, either using soft-label or hard-label, the model cannot achieve good performance, although CE loss with soft-label outperforms the FixMatch baseline (19.15%).
>
> This is because super-classes contain multiple categories, and it would be entirely wrong to treat them as labels to supervise a single sample.  As we discussed in our paper, it is challenging to provide accurate discriminative guidance on unlabeled data under the BSL setting. To this end, we exploit the relation (similarity and dissimilarity) between each unlabeled instance and the super-classes for guidance, and no label-based supervision was applied to the samples. This is 1) why we adopt a contrastive-like loss to provide such robust discriminative guidance and 2) the key to differentiate our model from other SSL methods that mostly focus only on prediction consistency.
>
> ## Q2: traning speed of K-Means.
> Since we apply clustering at each iteration that includes only 448 samples within the mini-batch, the convergence rate of the clustering procedure is quite fast. Besides, we implemented the K-Means algorithm on the GPU and set the maximum iterations of K-Means to 500. When running on a single 2080 Ti GPU, FixMatch takes 175s per epoch on average, while our method takes 190s per epoch on average. Thus our discriminative learning module only takes about **15s more seconds per epoch (8.57%)** and won't considerably slow down the training.
>
> ## Q3: random seed.
> The random seed indeed affects test performance, especially in BSL settings. In our paper, we run each experiment on five different random seeds: 1,2,3,4,5, which is also a common practice as most SSL methods FixMatch, CoMatch did. In table 2 in our paper, we compare our method with FixMatch for each random seed on CIFAR-10 with 10 labels. As discussed in FixMatch, it can sometimes achieve very high accuracy (85.11 seed=2), but sometimes obtain extremely low performance (19.15 seed=1 and 17.09 seed=4). On the contrary, our method can consistently outperform FixMatch and consistently achieves high accuracy on different seeds.
>
> ## Q4: datasets.
> Although the background of the images in the svhn dataset is noisy, current semi-supervised methods can all achieve very high accuracy on svhn. Meanwhile, as suggested, we also test our method on SVHN with 10 and 20 labels. Setting random seed=1, FixMatch can achieve high accuracies of 95.55% and 95.60%, respectively, while our method can obtain accuracies of 95.96% and 96.01%, respectively.
>
> On the other hand, in the experiments, we examine the performance of our method on the most commonly used datasets in SSL studies, CIFAR-10, CIFAR-100 (more classes), and STL-10 (larger image resolution). CIFAR-100 provides 50,000 images with 100 classes, which is comparable to Mini-ImageNet (60,000 images with 100 classes). And the size of the training images in the STL-10 dataset is usually 96×96, which is larger than that of Mini-ImageNet (84×84). We are sorry that we do not have sufficient time to provide additional test results on (Mini)-ImageNet during this short rebuttal period.

---

### Official Review · Reviewer_B558 · 2022-07-09

**Rating:** 6
**Confidence:** 3
**Soundness:** 3 good
**Presentation:** 2 fair
**Contribution:** 3 good

**Summary:**

This paper addresses Barely Supervised Learning (BSL), an SSL problem when only a few labeled samples are available.
The key is the introduction of a new loss function to the general consistency-based framework of SSL; the new loss measures the consistency between unlabeled samples and the centroids of "superclasses," which are obtained by applying K-means to the unlabeled data with $K <=$ # classes. K is gradually increased as the learning progresses, like a “curriculum learning” idea.
Experiments with CIFAR-10, CIFAR-100, and STL-10 have shown that the proposed method outperforms the existing SoTA SSL/BSL methods under BSL settings.


**Questions:**

There are several unclear points. In particular, I would like the authors to clarify C1-C3.

**Limitations:**

I did not find any discussion on limitations. Related to C2 above, if the training time increases when running K-means on a batch basis, it would be a clear weakness.

**Strengths And Weaknesses:**

--------------------------------------Pros--------------------------------------

P1. The paper addresses BSL, an SSL problem with only a few labels. While it is important and practical, few studies have been reported in the literature.

P2. This paper provides a good insight that BSL can be improved by a very simple method of adding consistency loss with K-means centroids.

P3. The method outperforms SoTA SSL/BSL methods.

P4. Ablation study on $K$ is reported.

--------------------------------------Cons--------------------------------------

C1. Batch size and $K$: Algorithm 1 says that K-means is performed batch by batch. Just to make sure, does this mean that K-means is applied to the samples within a batch, i.e., 64 samples? If yes, I wonder the resulting clusters will be quite different between iterations and may not be meaningful when the number of clusters is large.

C2. Computation Time: Since every iteration K-means is performed, the training time may increase. There is no discussion on this point.

C3. Results: I have some questions about the results. For example, in the CIFAR-100 100-label setting in Table 1, the score of LESS [4] is 28.2 $\pm$ 3.0, which is consistent with the number reported in [4]. However, in the 200-label setting, there is a gap between the numbers reported in this paper (39.5 $\pm$ 3.2) and those given in the original [4] (42.5 $\pm$ 3.2). It is unclear what is causing this difference. Such inconsistencies are also observed in Table 3.

C4. Analysis: The ablation study on the proposed loss term $L_{dis}$ would be essential, as it is the main idea of this paper. Some sensitivity analysis and discussions on the weight for $L_{dis}$, $\lambda_{dis}$, would be necessary.

C5. Novelty: The method looks a little incremental. The proposed method only adds the consistency loss with K-means centroids to the typical SSL framework (i.e., consistency between weak and strong augmented samples).

C6. The discussion of immutability and separability given in the introduction is interesting, but its significance is not clear. The relationship to the idea of the proposed method seems somewhat tenuous, and these measures are not used in the evaluations.

C7. Typos：
- L37 and others: “model collapse” -> "mode collapse" would be better.
- L49: “two different samples” -> "two samples from different classes" would be what the authors want to mean.
- L140 and others: “super-class super-class” -> “super-class”
- L190: “the consistent information learned model” -> "consistent information learning module" (the same is applied to L192: “the discriminative information learned model”)
- L221: “52.52%” -> “76.76%”

---

> ### Author Response · Authors · 2022-08-02
> **Response to Reviewer B558 (Q4-Q7)**
>
> ## Q4: discussion on loss weight $\lambda_{dis}$
> As suggested, we conducted experiments with the same seed and the results are shown below.
>
> |loss weight $\lambda_{dis}$|0.5|0.7|1.0|1.5|2.0|
> |-|:-:|:-:|:-:|:-:|:-:|
> |accuracy|80.64|80.95|81.28|79.54|78.90|
>
> Table R3-c. Results of different $\lambda_{dis}$. (seed = 1)
>
> Our model is insensitive to loss weights. In fact, the values of discriminative distribution loss and consistency loss are very close in order of magnitude, both  between 0.1 and 0.3. So we set the weight to 1.0.
>
> ## Q5: difference from consistency-based SSL methods
> Indeed, the method we propose is simple and easy to implement. Here, we would like to elaborate that we are **not simply adds the consistency loss with K-means centroids, but encourage the model to learn relative relations: samples are similar to their corresponding super-classes and not similar to other super-classes.**
>
> To our knowledge, **we are the first work to analyse the BSL problem from the perspective of immutability and separability, and to pioneer the solution of separability to improve the performance of SSL models under BSL.**
> As analysed in our paper, under BSL, semi-supervised models can easily produce model collapse due to insufficient discriminative information, and thus fail to distinguish between different categroies of samples. The similarity information we learn is a simple but reliable information that can be used as a complement to the discriminative information, thus solving the dilemma of semi-supervised models under BSL. The experimental results demonstrate the significant improvement of our proposed method for semi-supervised models under BSL.
>
> ## Q6: discussion of immutability and separability
>
> We propose immutability and separability to better understand the reasons for the failure of semi-supervised models in BSL. In our model, we use two types of loss to encourage the model being immutability and separability.
>
> - immutability--learning consistency loss
> Immutability is obtained in the classical SSL framework by learning consistency information between strongly and weakly augmented images, so do our model.
>
> - separability--learning discriminative loss
> As we mentioned in the previous question, the key to improving SSL models is to learn discriminative information so that they maintain separability even under BSL. We use clustering to learn the similarity/dissimilarity relationship between samples and super-classes in order to learn discriminative information so that the model keeps separability.
>
> Unfortunately, as immutability and separability are only the insights we propose, there is no standard way of measuring them. We propose the following approach for measurement. First, 10 images were selected from each category in the test set, for a total of 100 images.
>
> - evaluate immutability
>
> The 100 images were randomly perturbed and the model was tested to see if the predictions of these perturbed images were consistent with the original images. After testing (the model obtained after training on 10 CIFAR-10 labels), FixMatch and our model were correct at 100% and 98% respectively, which indicates that the immutability of the model can be satisfied under BSL.
>
> - evaluate separability
>
> We calculate the confusion matrix for the predicted results of these 100 images.
>
> |class index|1|2|3|4|5|6|7|8|9|10
> |-|:-:|:-:|:-:|:-:|:-:|:-:|:-:|:-:|:-:|:-:|
> |1|10|0|0|0|0|0|0|0|0|0|
> |2|8|1|0|0|0|0|0|1|0|0|
> |3|10|0|0|0|0|0|0|0|0|0|
> |4|8|0|0|2|0|0|0|0|0|0|
> |5|7|0|0|0|2|1|0|0|0|0|
> |6|10|0|0|0|0|0|0|0|0|0|
> |7|10|0|0|0|0|0|0|0|0|0|
> |8|9|0|0|0|0|0|0|1|0|0|
> |9|7|1|0|0|0|0|0|0|2|0|
> |10|9|0|0|0|0|0|0|0|0|1|
>
> Table R3-d. FixMatch's confusion matrix (10 labels on CIFAR-10 with seed=1)
>
> |class index|1|2|3|4|5|6|7|8|9|10
> |-|:-:|:-:|:-:|:-:|:-:|:-:|:-:|:-:|:-:|:-:|
> |1|9|0|0|0|0|0|0|0|1|0|
> |2|0|10|0|0|0|0|0|0|0|0|
> |3|0|0|7|0|0|1|2|0|0|0|
> |4|1|0|0|7|0|2|0|0|0|0|
> |5|2|0|0|0|8|0|0|0|0|0|
> |6|0|0|0|0|0|7|0|3|0|0|
> |7|0|0|1|0|0|0|9|0|0|0|
> |8|0|0|0|0|0|0|0|10|0|0|
> |9|4|0|0|0|0|0|0|0|6|0|
> |10|0|1|0|0|0|0|0|0|0|9|
>
> Table R3-e. Our model's confusion matrix (10 labels on CIFAR-10 with seed=1)
>
> Separability refers to the ability of the model to distinguish between different classes of images. Obviously, the confusion matrix of our model indicates better discriminative power (i.e., separability) of our model with small amount of mis-classification shown on the off-diagonal parts.
>
> ## Q7: typos
> Thank you for pointing out the Typos. We will check our article carefully and make changes in subsequent versions.

---

> ### Author Response · Authors · 2022-08-02
> **Response to reviewer B558 (Q1-Q3)**
>
> ## Q1: the resulting clusters
>
> Sorry to cause the confusion. Here 64 is just the number of labelled images in a batch, while each iteration involves 7 times the proportion of unlabelled samples (following FixMatch,FlexMatch). Therefore each clustering actually clusters 448 (64x7) images.
>
> In addition, we evaluate how the clusters (super-classes) are formed with regards to object categories with the progression of training in Table R3-a. The values (1-10) in the table correspond to object categories forming a super-class.
>
> ||super-class A|super-class B|super-class C|super-class D|super-class E
> |-|:-:|:-:|:-:|:-:|:-:|
> |100,000th iteration|8|2,5|3,4,6,7|1,9|10|
> |150,000th iteration|1,9|4,6,7|2,10|5,8|3|
> |200,000th iteration|1,9|8|2,10|5|3,4,6,7|
>
> Table R3-a. Composition of super-classes. Here we want to explain clearly the *"composition of the super-class"*. As shown in Table R3-b, the first row indicates the percentage of samples from category 1 that were assigned to each super-class, and since 91.5% of the samples were assigned to super-class D, we say that category 1 is the composition of super-class D.
>
> |Category|super-class A|super-class B|super-class C|super-class D|super-class E|max(super-class)
> |-|:-:|:-:|:-:|:-:|:-:|:-:|
> |1|0.0%|6.0%|0.0%|91.5%|2.5%|D
> |2|1.4%|86.4%|0.0%|1.2%|0.0%|B
> |3|5.6%|0.0%|94.4%|0.0%|0.0%|C
>
> Table R3-b. Percentage of samples assigned to each super-class
>
> We first show that the clustering results don't fluctuate significantly. Comparing the results of 100,000 iters with that of 200,000 iters, only the clustering results for categories 2,5,10 have changed. Thus the relative relations of majority unlabeled samples is not changing a lot.
>
> As a result, **such dynamic clustering results actually have less impact on our approach, since it is the relative relation between each sample and the super-classes that provides the discriminative information to guide our learning.** Specifically, instead of enforcing the prediction of unlabeled sample to a certain category, we exploit the relative relations to discriminating the unlabeled sample to be similar to the distribution of its corresponding super-class compared to that of others. Such robust strategy is the key in our method to provide effective discriminative information.
>
> ## Q2:  computation time
>
> Since we apply clustering at each iteration that includes only 448 samples within the mini-batch, the convergence rate of the clustering procedure is quite fast. Besides, we implemented the K-Means algorithm on the GPU and set the maximum iterations of K-Means to 500. When running on a single 2080 Ti GPU, FixMatch takes 175s per epoch on average, while our method takes 190s per epoch on average. Thus our discriminative learning module only takes about **15 more seconds per epoch (8.57%)** and won't considerably slow down the training.
>
> ## Q3: the results
>
> Sorry to cause the confusion. For the results reported in Table 3, since LESS did not report the results for the very few labels on STL-10 and 20 labels on CIFAR-10, we have to re-run their code to produce these results for comparison. For the results reported in Table 1, in terms of the 200-label setting on CIFAR-100, we reproduce the result of LESS as 39.5±3.2 using its released code, while the original paper gave the results as 42.5±3.2, both of which are inferior to our result (45.62±1.39). Whereas for the results of 250 labels on CIFAR-10, table 2 in the original paper reported 95.0 ± 0.8 and table 3 reported 95.1 ± 0.8, we adopted the results of Table 3.

---

### Official Review · Reviewer_5pm6 · 2022-07-10

**Rating:** 6
**Confidence:** 4
**Soundness:** 3 good
**Presentation:** 3 good
**Contribution:** 3 good

**Summary:**

The paper tackles the problem of barely-supervised learning – an SSL setting in which only a few-labels per class are annotated. The authors propose a novel approach that extends the FixMatch with a discriminative information learning module. The key idea of this module is to form super-classes by clustering the data using K-means clustering and then rely on the similarity between super-classes and samples to guide the training. The number of clusters is gradually increased during the training. In that way, super-classes are dynamically formed and gradually become more fine-grained. The final objective function consists of standard cross-entropy loss on the labeled examples, standard consistency loss based on the weakly and strongly augmented versions of the data and the newly introduced discriminative distribution loss based on the formed super-classes. The performance of the approach is compared to state-of-the-art SSL and barely-supervised methods on CIFAR-10, CIFAR-100 and STL-10 datasets.

**Questions:**

- Why K ∈ {5,10,20} for CIFAR-100? What is the performance on CIFAR-100 when the K goes to the maximum of 100 classes?
- Why is K increased only in the 30% of iterations? How is that determined and how does a different setup affect the performance?
- How does the algorithm compare to others on the SVHN benchmark dataset on the barely-supervised learning as well as standard SSL setting?
- How does the algorithm perform on the unbalanced dataset?
- Authors motivate super-class idea by the fact that the pseudo-labelling approach can be very noisy. But how accurate are the resulting clusters (super-classes) during model training? It would be beneficial to compare the accuracy of pseudo-labels of FixMatch with the the quality of generated clusters in the proposed approach (e.g., whether examples that belong to the same ground-truth class are assigned to the same super-class during training)

**Minor comments:**
- Line 194: it seems it should be \lambda_con=\lambda_dis=1 not L
- Line 294: sentence is not finished


**Limitations:**

No. I suggest authors comment on the limitations of their reliance on the K-means clustering.

**Strengths And Weaknesses:**

**Strengths:**
- Paper is well written, well structured and it was a pleasure to read it. The explanation with immutability and separability motivates the paper and the proposed approach really well.
- Barely-supervised learning is a challenging setting, and the proposed method solves the challenge in a simple, elegant and effective way
- The proposed  method achieves state-of-the art performance on the barely-supervised learning setting on three datasets, and comparable performance on the standard SSL setting.

**Weaknesses:**
- (Transductive) few-shot learning should be discussed in the related work and in the introduction and the differences between the settings need to be explained. Barely supervised learning should be better defined in the introduction and put in the relation with the more studied problem of the few-shot learning.
- The performance does depend on K and it is not clear how to set it. For example, it is not clear why for the CIFAR-100 K was set in the range {5,10,20} and differently then for the CIFAR-10 and STL-10 datasets. How does setting K differently affect performance on the more challenging CIFAR-100 dataset? The analysis is only done on the CIFAR-10 dataset.
- Reliance on the K-means clustering. Clusters can only be sperical in shape and the method may fail for the unbalanced dataset.

---

> ### Author Response · Authors · 2022-08-02
> **Response to reviewer 5pm6 (Q5)**
>
> ## Q5: discussion on the accurate of the resulting clusters (super-classes)
> As the samples do not have a fixed super-class attribute, we first counted the distribution of each category of sample to assess the reliability of our clustering. The Table R2-d shows the clustering distribution at the 50,000th iteration (10 labels on CIFAR-10), where each row represents the percentage of samples in one category that were assigned to each super-class.
>
> ||super-class A|super-class B|super-class C|Max|
> |--|:---------:|:-----------:|:-----------:|:-:|
> |1 |0.00%	  |84.44%	    |15.56%       |84.44%|
> |2 |92.86%	  |4.76%        |2.38%        |92.86%|
> |3 |0.00%	  |45.10%	 	|54.90%		  |54.90%|
> |4 |0.00%	  |8.11%	    |91.90%		  |91.90%|
> |5 |0.00%	  |0.00%		|100.00%	  |100.00%|
> |6 |0.00%	  |3.51%		|96.49%   	  |96.49%|
> |7 |0.00%	  |0.00%		|100.00%	  |100.00%|
> |8 |0.00%	  |5.00% 	    |95.00%		  |95.00%|
> |9 |0.00%	  |100.00%		|0.00%		  |100.00%|
> |10|87.76%	  |8.16%		|4.08%		  |87.76%|
>
> Table R2-d. The clustering distribution at the 50,000th iteration
>
> The "MAX" column indicates *up to how many samples that belong to the same ground-truth class are assigned to the same super-class.* If we count samples of the same ground-truth class and being in the same super-class as the correctly clustered samples, we can obtain the clustering average accuracy at the 50,000th iteration above as 90.34%.
>
> More importantly, as suggested, we compare the accuracy of pseudo-labels generated by FixMatch and our methods over a larger number of iterations. (we set the random seeds to 1 and 2 respectively, as shown in the following two tables)
>
> |#iterations|3w|5w|10w|20w|30w|40w|50w|100w|test-best-acc
> |-|:-:|:-:|:-:|:-:|:-:|:-:|:-:|:-:|:-:|
> |FixMatch  |17.1%|12.0%|10.2%|11.7%|9.1% |13.6%|8.6%|9.8%|19.15%|
> |ours      |34.3%|37.1%|63.6%|74.8%|79.1%|80.7%|82.6%|85.0%|81.28%
> |clustering|77.0%|90.3%|87.5%|90.8%|91.7%|85.2%|85.6%|87.3%|-
>
> |#iterations|3w|5w|10w|20w|30w|40w|50w|100w|test-best-acc
> |-|:-:|:-:|:-:|:-:|:-:|:-:|:-:|:-:|:-:|
> |FixMatch  |22.8%|26.9%|33.1%|46.1%|55.4%|79.5%|80.1%|86.3%|85.11%|
> |ours      |37.1%|47.0%|59.1%|64.3%|80.1%|82.4%|87.4%|88.9%|86.12%|
> |clustering|76.0%|92.4%|90.4%|91.6%|91.5%|89.6%|90.8%|90.8%|-
>
> Table R2-e.  The rows "FixMatch" and "ours" indicate the accuracy of the pseudo-labels at different iterations of FixMatch and our method, respectively. The line "clutering" indicates the accuracy of our model in clustering (K=3 in 3w,5w iteration, K=5 in 10w,20w,30w iteration, K=10 in 40w,50w,100w iteration).
>
> It is clear to see that the super-classes we have implemented are more reliable compared to pseudo-labelling. Especially at the early stage of training, the accuracy of pseudo-labels is very low, so we perform clustering of super-classes, by which simple super-classes can learn more accurate information, thus helping model training.
>
> ## More discussion
> ### few-shot learning and BSL
> In few-shot (K-shot) setting，there are a large training set of base classes, a small support set of novel classes and a query set. When K=1, there is only one label per novel class, which seems very similar to BSL. However, we have to emphasize that the model does not learn only on the support set, but also on the large training set. Thus the model in the few-shot task is trained entirely on labeled samples, and unlabeled samples only participate in the test on the query set. In contrast, in BSL, the model is trained on a very small number of labeled samples and a large number of unlabeled samples. Therefore, BSL and the few-shot learning task are completely different learning tasks.
>
> ### typos
> Thank you for pointing out the error. In line 194, as you say, it should be $ \lambda_{con}=\lambda_{dis}=1$; In line 294, the complete sentence is "the test performance achieve the best when $\tau_2$ =0.8". We will revise it in a later version.

---

> ### Author Response · Authors · 2022-08-02
> **Response to reviewer 5pm6 (Q1-Q4)**
>
> ## Q1: the range of K settings
> The choice of K is limited by clustering algorithm performed at each training iteration. More specifically, we perform the clustering algorithm in each training iteration, i.e., there are only 448 unlabeled samples per clustering. If we divide them into 100 clusters with an average of only 4 samples per cluster, the super-class representation can be very noisy. Our experimental result at K=100 is 27.4% (10% lower than our setting). Therefore, we set K ∈ {5,10,20}, which is a reasonable range, taking into account the clustering algorithm and the sample size requirements.
>
> ## Q2: the growth rate of K
> K is increased only in the 30% of iterations (i.e., K’s growth rate $\alpha$ = 0.3). We further evaluate the influence of different values of $\alpha$, i.e., the growth rate of K, in Table R2-a.
>
> |growth rate $\alpha$|20%|30%|40%|50%|
> |-|:-:|:-:|:-:|:-:|
> |accuracy|76.84 ± 8.51 | 78.25 ± 8.08 | 78.59 ± 9.68 | 76.78 ± 8.33|
>
> Table R2-a. The results of different growth rates of K (under 3 random seeds)
>
> As can be seen, overall, the growth rate $\alpha$ of K is not significant to model performance, as long as $\alpha$ is in an appropriate range.
>
> ## Q3: performance on the SVHN dataset
> The experiments on SVHN dataset are shown in the following table.
>
> ||10 labels|20 labels|40 labels
> |-|:-:|:-:|:-:|
> |FixMatch|95.55|95.60|95.98|
> |ours|95.96|96.01|96.14|
>
> Table R2-b. Results for the SVHN dataset.
>
> SVHN is a relatively simple SSL dataset as most current SSL methods can easily achieve an accuracy above 95%. It can be seen that FixMatch has been able to achieve very good performance with very few labels on this dataset. SVHN is therefore not optimal to demonstrate the benefits of our approach. Instead, in our paper we have evaluated our methods on more challenging datasets under BSL.
>
> ## Q4: unbalanced dataset
> Thank you for pointing out the limitations of K-Means. To explore the problem in an unbalanced dataset, we apply unbalanced sampling to the unlabeled samples on CIFAR-10, together with either 10 or 40 balanced labeled samples, respectively. Specifically, we sampled the unlabeled data from the 10 categories with the ratio of 1:2:3:4:5:6:7:8:9:10.
>
> ||10 labels|40 labels
> |-|:-:|:-:
> |FixMatch|18.85|71.28
> |ours|62.43|88.46
>
> Table R2-c. Results in unbalanced scenarios.
>
> As can be seen, unbalanced data is an extremely challenging scenario.  With 10 labels, our method achieves 81.28% when the categories are balanced, but in the imbalance scenario, our model performance is reduced by nearly 20%. Of course, our method is not completely ineffective and can still be a little better than baseline (FixMatch).
>
> The influence of the limitation of K-means on our method is further analyzed as follows. The imbalance condition makes it easy for smaller categories to be drawn into the larger category to form a cluster. However,  this may not affect our method significantly as expected since our super-classes themselves by nature contain several categories and they are therefore less affected by this limitation of K-means when the number of super-classes is small. However, as the number of super-classes gradually increases, the clustering results may become progressively noisier due to this limitation, and the learning of the inference model will be more affected.
>
> We will look into it further in our subsequent work. Thank you again.

---

### Official Review · Reviewer_FJpN · 2022-07-10

**Rating:** 6
**Confidence:** 3
**Soundness:** 3 good
**Presentation:** 3 good
**Contribution:** 2 fair

**Summary:**

From the notion and empirical analyses from barely supervised learning, it introduces super class based regularization which improves the discriminability of unlabeled data instances. It largely improves the performances with various benchmark datasets with high confidences. The introduction of clustered super class is somewhat novel in the semi-supervised learning.

**Questions:**

Stated in weaknesses session.

**Limitations:**

Stated in weakness session.

**Strengths And Weaknesses:**

Strengths :

Introduction of super class is the main novelty of this paper. It mitigates the imbalanced ability of barely supervised learning from the side of immutability and separability, which is sound.

Weaknesses :

Application on the semi-supervised learning with more labels provides ineffective performances, which is not explainable from the view of immutability and separability. If it really matters, the performances can be calibrated from the hyper-parameter tuning between consistent learning and the introduced discriminative learning.

---

> ### Author Response · Authors · 2022-08-02
> **Response to reviewer FJpN**
>
> Thanks for raising this discussion. We first clarify the immutability and separability. As we presented in the paper, most SSL methods focus only on the models’ immutability, i.e., enforcing the prediction consistency from two different augmented variants of the same image. These methods rely totally on the labeled samples to learn separability, i.e. discriminating an instance from others. However, under the BSL scenarios where the labeled data is scarce, these methods cannot achieve promising results due to losing the separability, leading to model collapse that all the instances are predicted into a single or few classes. Our method carefully explored the discriminating guidance for SSL training, achieving a remarkable performance improvement under BSL scenarios.
>
> **Certainly, when there are plenty of labeled samples, labeled (correct) information is the most effective guidance to achieve good discriminating ability.** We can observe from the table 3 in our paper, that all different SSL methods can achieve very high and close test accuracy when sufficient labels are provided (like 250 or 4000 labels on CIFAR10) so that both immutability and separability could be effectively catered for.
>
> ||10 labels | 20 labels | 40 labels|250 labels|
> |--|:--:|:---:|:-:|:-:|
> |FixMatch| 44.77 ± 24.99 | 80.46 ± 5.15 | 90.11 ± 3.01 | 94.21 ± 0.61 |
> |Ours    | 76.76 ± 6.78  | 88.49 ± 3.26 | 94.19 ± 0.41 | 94.54 ± 0.27 |
>
> Table R1-a. Comparison of our method with FixMatch (baseline)
>
> **It is also worth mentioning that our method is used on top of the FixMatch base framework.** We can observe from the Table R1-a that our method can outperform our baseline (FixMatch) even when there are more labels available, though the improvement is not significant.
>
> On the other hand, as you suggested, we also carefully fine-tune the hyper-parameters of our method to check whether the method can obtain higher accuracy when more labels are available. In Table R1-b, we test the performance of our baseline (FixMatch) and our method on CIFAR-10 with 40 labels, and 250 labels, respectively, under the same experiential settings (random seed=1). Specifically, when there are more labeled samples carrying precise discriminative information, we reduce α, the growth rate of K to 0.1 and the loss weight $λ_{dis}$ to 0.3 to mitigate the influence of the additional supervision by our proposed discriminative distribution loss. As shown in Table R1-b, this strategy could further improve the performance of our model under more label settings. In our paper, to make the experiment consistent, we did not  fine-tune hyper-parameters but applied K =0.3 and $λ_{dis}$=1.0 consistently to all experiments.
>
> ||40 labels|250 labels
> |--|:--:|:---:|
> |FixMatch| 92.64 |94.36  |
> |Ours    | 93.90 |94.49  |
> |Ours ( tuning hyper-parameter)|94.05|94.57|
>
> Table R1-b. Results after tuning hyper-parameters (seed=1)

---

> > ### Comment · Reviewer_FJpN · 2022-08-04
> > **Thanks for your rebuttal.**
> >
> > From the results with fine-tuned hyper-parameters, i could better understand the main differences between semi-supervised learning and barely-supervised learning. Also, the proposed method shows robust performances across the difference on the number of labeled samples. I will increase my score from 5 to 6.
> >
> > Can i ask you for further results on hyper-parameter tuning results on low-sample experiments? (i.e. 10 samples or 1 samples). I am more than eager to increase my score to 7 when i get convinced about the tuning result.

---

> > > ### Author Response · Authors · 2022-08-08
> > > **More results**
> > >
> > > Thank you for your interests. We applied the new-parameters used in rebuttal ($\alpha$=0.1,$\lambda_{dis}$=0.3) to low-label cases (i.e., only one or two labeled samples per class). The experimental results are as follows ("default-parameters” means $\alpha$=0.3,$\lambda_{dis}$=1.0 that was adopted in our paper).
> > >
> > > |# labels|10 labels|20 labels|40 labels|250 labels|
> > > |-|:-:|:-:|:-:|:-:|
> > > |FixMatch|19.15|81.50|92.64|94.36
> > > |ours (default-parameters)|**81.28**|**90.02**|93.90|94.49|
> > > |ours (new-parameters)|76.64|89.56|**94.05**|**94.57**|
> > >
> > > Table R1-c. Results with default-parameters and new-parameters (seed=1)
> > >
> > > We can see that the default-parameters that emphasize more on the discriminating model are more appropriate for the BSL scenario (10 or 20 labels) and less effective for the SSL scenarios with more labels. Just as discussed in our paper, the BSL setting without sufficient labeled information requires more additional discriminating information to guide the learning. On the other hand, the new-parameters that weaken our discriminating model  are more suitable for SSL scenarios when  more labels carrying precise discriminative information have already been available. Though under BSL setting the test performance of our method is degraded with new-parameters, our method can still outperform the baseline by a large margin.
> > >
> > > Besides, we also provided more ablation studies on the influences of hyper-parameters in Sec 3.4. By default, we adopt $\alpha$=0.3,$\lambda_{dis}$=1.0 to run to all experiments.

---

### Author Response · Authors · 2022-08-09
**Looking forward to more discussions**

Dear Reviewers,

Thanks again for your valuable comments! We have carefully responded to all your concerns.

 In our work, we argue the two main factors affecting the performance of BSL/SSL, immutability and separability. Especially, we highlight the importance of discriminative information for BSL scenarios that is critical but ignored by most existing SSL methods. To this end, we explored the super-classes dynamically to provide appropriate discriminating guidance, i.e.,using the relative relations to enforce the unlabeled sample to be closer to its corresponding super-class than to other super-classes. As a result, our methods can achieve significant and consistent improvement in BSL without deteriorating SSL with more labels.

We are looking forward to your feedback and are eager to discuss more with you.

Thank you, authors

---

### Meta-Review · Area_Chair_RQsX · 2022-08-27

**Recommendation:** Accept
**Confidence:** Certain

**Metareview:**

This paper proposes a simple but effective method for barely-supervised learning, where the label is scarce, for instance, 1 example per class. The method is based on k-means clustering of the unlabeled data to form superclass and rely on a contrastive-type of loss to enforce discriminativeness. The proposed method improves class separation in the presence of scarce labels and significantly improves the overall performance.

Though the proposed method is rather heuristic, the idea to address barely supervised learning is novel and interesting.  The superior performance is demonstrated via strong empirical studies. I would recommend acceptance of this paper given the novelty and simplicity of the idea and the strong empirical evidence.


**Award:**

No

---

### Decision · Program_Chairs · 2022-09-14

Accept